# Parents’ Personality, Marriage Satisfaction, Stress, and Punishment of Children in the Family

**DOI:** 10.3390/bs9120153

**Published:** 2019-12-12

**Authors:** Mikhail Chumakov, Daria Chumakova

**Affiliations:** Department of Psychology, Kurgan State University, 640020 Kurgan, Russia; ch-darya@mail.ru

**Keywords:** parents’ personality, punishment of children, stress, satisfaction

## Abstract

The objective of the research is to ascertain whether emotional and volitional characteristics of the individual affect the frequency of punishment in the family (for example, prohibition to watch TY for some time, pocket money reduction, request for an apology). These types of punishments are legal in Russia. The research method is Perrez’s, Schoebi’s questionnaire on punishing behavior in education, and Chumakov’s questionnaire on volitional and emotional characteristics of a person (VEC). The data were divided into two groups (high or low level of development of volitional powers). Analysis of variance (ANOVA) was applied to compare groups. The dependent variables were marriage satisfaction, emotional stress, and frequency of punishment of children in the family. Results and conclusions found that the gaps between the groups were significant for marriage satisfaction F (1,105) = 17.77, *p* < 0.001 and emotional stress F (1,105) = 23.53, *p* < 0.001 but were not significant for the frequency of punishment. Marriage satisfaction in a group with high VEC parameters is higher than in the group with low parameters. The stress in the group with high VEC parameters is lower than in the group with low parameters. There is a correlation between marriage satisfaction and frequency of punishment as well as between stress and frequency of punishment. Thus, the emotional and volitional qualities of the parents’ personality indirectly influence the frequency of punishment of children in the family.

## 1. Introduction

A number of researches regard stress in connection with family relations [1]. In particular, the interconnection among stress experienced by the parents, family life satisfaction, and the punishment of children is currently under discussion. Research and studies, including cross-cultural ones, show that high marriage satisfaction is connected with low stress levels, and the increase of the stress level is connected with the higher frequency of the punishment of children, including corporal punishment [2,3,4,5,6,7]. Previous research has focused on corporal punishment and characteristics of family relations, including stress. Our research focuses on the parents’ personal traits in connection with the characteristics of the family relations such as marriage satisfaction, stress level of the spouses, and frequency of punishments of children in the family. The personal traits analyzed in the research include volition and emotional stability. The hypothesis of the interconnection between volition and emotional stability as well as low frequency of punishment of children was to be verified. Therefore, the presented research accomplishes the analysis of the interconnection of parents’ personal traits, family relations, as well as frequency and type of punishment of children.

This study of child punishment by their parents proceeds from the fact that there are personality traits in adults that contribute to an increase in punishment frequency and personality traits that contribute to a decrease in their frequency. Additionally, studies on physical punishment have shown the role of low socio-economic status, frustration, and some cultural norms in their frequent use [4]. Our study is devoted neither to socio-economic factors nor cultural characteristics, but to personal traits of parents in connection with the problem of child punishment in the family. This study has been approved by the ethics committee of the Kurgan State University.

## 2. Materials and Methods

The participants of the research were 280 adults, all of which have children under the age of 16. The research was carried out between 2017 and 2018. Out of 280 people, 107 people were selected; they were then divided into two groups. Criteria of selection—High (8, 9, 10 of standard ten) and low (1, 2, 3 of standard ten) test scores. The first group included 43 people showing low results in the test by Chumakov (2006) [8]. The second group included 64 people with high results in the test by Chumakov. The first group consisted of 14 men and 29 women. The average number of children in the family was 1.67 (SD = 0.81). The average age of the children was 8.86 (SD = 4.59). Among the children, there were 19 boys and 24 girls. The second group consisted of 45 men and 48 women. The average number of children was 1.63 (SD = 0.83). The average age of the children was 8.28 (SD = 4.85). There were 29 boys and 35 girls in the group. Therefore, taking into consideration the above mentioned criteria, the groups were homogeneous, a condition suitable for experimental research [9,10]. Analysis of variance (ANOVA) was applied to compare the groups [11]. The dependent variables were marriage satisfaction, emotional stress, and frequency of punishment of children in family. All the parents completed a number of tests, measuring various aspects of punishment of children, the stress experienced by people in family life, family life satisfaction, as well as volitional and emotional personal traits. Social desirability was measured using the scale SDS CM (Crowne and Marlowe, 1960). The frequency of punishment was measured with the scale of the parents’ punishment frequency created by Deneker (1988) and completed by Perrez, Ewert, Moggy (1992). The scales are a system for estimating occurrences of applying various punishments to children and comprises two parts. The first part consists of a scale containing 12 questions that reflect the frequency of child punishment in the family and the answers ranging from 1 to 6 (from “almost never” to “very often”). The 12 questions contain 12 types of punishments, the frequency of which parents should choose considering their actual behavior. In addition, the subjects completed this test (i.e., filled out this scale) again with respect to the frequency with which these types of punishments were applied to them in their parents’ family. The second part consisted of an assessment by parents of how frequently they applied 10 punishments in eight different situations of a child’s behavior in the range from 1 to 6 (from “yes, surely” to “surely not”). Thus, the subject assessed the frequency of each of the 10 types of punishment in eight instances, taking into consideration the child’s behavior in various situations.

Emotional stress and family life satisfaction were measured by means of the scale of the emotional stress measurement and the scale of satisfaction with various spheres of family life created by Perrez, Schoebi, Wilhelm (2000). The scale of the emotional stress measurement includes seven questions related to the frequency of the emotional expression in everyday life (variants of answers from 1 to 4, from “very seldom” to “very often”). The scale of satisfaction with various spheres of family life includes five questions (variants of answers from 1 to 6, from “very unsatisfied “to“ absolutely satisfied”). The scale Depressed Mood, created by Perrez and Schoebi, consists of the direct estimation of depression resulting from household duties and family life (from 1 to 6, from “not depressed” to “extremely depressed”) [12]. The scales were translated into the Russian language and adapted by Chumakov (2004). Afterwards, the reverse translation was accomplished. The data consistency of the Russian version of the scales was satisfactory (Chronbach’s alpha from 0.72 to 0.74). Volitional traits and emotional stability were measured by means of the scale of volitional and emotional characteristics of a person by Chumakov (2006) [8]. The hypothesis of our research is that the more parents’ emotional and volitional characteristics are expressed, the less frequent are punishments administered.

## 3. Results

The gaps between the groups were significant in terms of marriage satisfaction F (1,105) = 17.77, *p* < 0.001 and emotional stress F (1,105) = 23.53, *p* < 0.0001 but were not significant for the frequency of punishment. For the depressed mood parameter, the differences were close to significant F (1,105) = 3.90, *p* = 0.51.

The stress level was higher in the first group (the group with low results in the test by Chumakov, M_1_ = 5.61, SD_1_ = 3.30; M_2_ = 2.91, SD_2_ = 2.45) (2006) [8]. Marriage satisfaction was higher in the second group (the group with low results in the test by Chumakov, M_1_ = 19.07, SD_1_ = 5.58; M_2_ = 23.22, SD_2_ = 4.56) (2006) [8]. For the depressed mood parameter, the differences were close to significant F (1,105) = 3.90, *p* = 0.051. The level of depressed mood was higher in the first group (M_1_ = 1.91, SD_1_ = 1.21; M_2_ = 1.48, SD_2_ = 0.99). The frequency of punishments was lower in the second group (M_1_ = 26.40, SD_1_ = 6.92; M_2_ = 25.31, SD_2_ = 7.23). Negative correlation was observed between marriage satisfaction and the frequency of punishments (r = −0.20, *p* < 0.01), but stress correlated with the frequency of punishments positively (r = 0.17, *p* < 0.01).

The study found VEC negative correlations with the frequency of two types of punishment application (r = −0.29, *p* < 0.01) and the frequency of eight types of punishment application (r = −0.21, *p* < 0.05) in one situation of child behavior. Negative correlations with the frequency of two types of punishment application (r = −0.25, *p* < 0.01) and the frequency of eight types of punishment application (r = −0.20, *p* < 0.05) in four situations of child behavior. Negative correlations with the frequency of two types of punishment application (r = −0.26, *p* < 0.01) in five situations of child behavior. Negative correlations with the frequency of two types of punishment application (r = −0.22, *p* < 0.05) and the frequency of eight types of punishment application (r = −0.20, *p* < 0.05) in seven situations of child behavior. A positive correlation was obtained between the index of frequency of punishments applied to children and the frequency of punishments applied to their own parents in their childhood years (r = 0.38, *p* < 0.01).

## 4. Discussion

Most research has been devoted to the interconnection between the frequency of punishments and stress level, family life satisfaction, and other aspects of family life. However, while these studies state the frequency of punishments, in this article, the objective laws and the determination of the frequency of punishments are under analysis. We have verified the hypothesis according to which the personal traits of the parents influence indirectly the frequency of punishments. They also influence the stress level and family life satisfaction, whereas these aspects are connected with the frequency of punishments in the family. Therefore, our research presents new data concerning the mechanisms formerly discovered. The positive correlation of the stress level to the frequency of punishments in the family has been already determined in cross-cultural research, as well as tendencies in the interconnection between personal traits and family relations [12,13]. Our data justify and complete the previously obtained results. As an addition to the obtained data, our research reveals the interconnection between the volitional traits of the parents and the stress level as well as their indirect influence upon the frequency of punishments in the family. The respondents with highly developed volition experience stress less frequently, which leads to the decrease in the frequency of punishments. In our research, we proceeded from an assumption that parents with a certain set of personality traits punish their children less often. Personality traits of parents influence the punishment occurrence indirectly through a higher satisfaction with family life and a low level of emotional stress of parents with a certain set of personality traits. Personal characteristics of parents were investigated using the VEC test as well as the parents’ role when explaining the results.

To explain the results obtained, let us consider in more detail what personality traits and behaviors the VEC test diagnoses. The research subjects with high test scores are self-confident, independent, able to control anger, active, persistent, and have a clear-cut goal in life. This set of characteristics alone cannot explain the low frequency of child punishment in the family, except for the ability to control anger. Probably, this parameter is one of the reasons for the difference between the experimental groups in terms of punishment frequency. The impulsive punishment under the influence of this emotion decreases in terms of number. Research shows that satisfaction with family life consists in satisfaction with the ability of the family to control stressful situations, satisfaction with the distribution of responsibilities in the family, satisfaction with how problems are discussed and decisions are made in the family, how family members take care of each other, and satisfaction with the climate and atmosphere that usually prevails in the family. The difference between the groups with high and low scores according to the VEC testing technique with regard to satisfaction with family life is confirmed and partially explained by the VEC indicators’ positive correlation with the family adjustment scale in the H. Bell Adjustment Inventory (r = 0.22, *p* < 0.05) obtained in our previous studies [12].

Satisfaction with responsibility distribution, which is characteristic of the people with high VEC indicators, can be explained by the fact that they are more efficient and active, including with regard to their performance of family responsibilities. As a result, such responsibilities feel like less of a burden and their distribution is concerned about less as well. Anger control has a positive effect on the family climate and atmosphere, which increases satisfaction.

The parents with high VEC indicators are better at managing emotional stress, which partly explains the higher satisfaction with managing stressful situations in the family. However, for a more complete understanding, further research should be carried out. This article states that personal and behavioral characteristics recorded by the VEC test affect satisfaction with family life. This fact, in its turn, may explain why the subjects with high VEC indicators punish their children less frequently, taking into consideration a negative correlation between satisfaction with family life and punishment frequency. Emotional stress considered in the study consists of the predominance and high frequency of emotions such as sadness, anxiety, guilt, embarrassment, and dissatisfaction in combination with a low frequency of emotions such as joy and self-satisfaction. The negative correlation of the indicator according to the VEC testing technique and of emotional stress can be explained by the content of personal and behavioral manifestations at high VEC scores. Active, confident, persistent people and those having a clear–cut goal in life have higher achievements and, therefore, more reasons for self-satisfaction and joy and fewer reasons for sadness, anxiety, guilt, embarrassment, and dissatisfaction. A negative correlation between the VEC indicator and emotional stress also explains why the subjects with high VEC scores punish children less frequently.

The correlation between the VEC method index and other personality traits will help to better understand how the volitional and emotional traits of parents influence the punishment of children in the family. Correlation indicators by the VEC test and NEO-PY-R R will explain the results.

In previous studies, we obtained VEC correlations with NEO PI-R R test scales such as neuroticism (N), (r = −0.48, *p* < 0.05) and extraversion (E), (r = 0.46, *p* < 0.05). In the neuroticism factor, the VEC correlates most strongly with the anxiety subscales (N 1), (r = −0.42, *p* < 0.05), self-consciousness (N 4) (r = −0.65, *p* < 0.01), and vulnerability to stress (N 6), (r = −0.42, *p* < 0.05). In the extraversion factor, the VEC most strongly correlates with the assertiveness subscale (E 3), (r = 0.54, *p* < 0.01) and activity subscale (E 4), (r = 0.68, *p* < 0.01). In the agreeableness factor, the VEC correlates with the trust subscale (A 1), (r = 0.41, *p* < 0.05) and in the conscientiousness factor with achievement-striving subscale (C 4), (r = 0.41, *p* < 0.05). Correlations with such subscales as depression (N 3) (r = −0.27) and impulsiveness (N 5) (r = −0.35) do not reach statistical significance; however, they can be considered a trend [9]. These data are consistent and complement the picture of personality traits diagnosed by the VEC test. Some of them, such as trust, help to understand the interrelation with the punishment of children in a family. In terms of the Big Five factors, one can give the following description of a person with high VEC indexes: Anxious, nervous, stressful, and apprehensive (N 1); Prone to feel guilty and sad and be in a depressed state (N 3); Prone to experiencing emotion of shame, sensitive to taunts, prone to feeling of inferiority (N 4); Unable to control desires and impulses (N 5); Sensitive to stress, less able to deal with stress, is stricken with panic when confronted with threatening situations (N 6); Not dominant or not trying to dominate other people (E 3); Not showing high activity (E 4); Not trusting other people (A 1); Satisfied with a low level of his achievements (C 4). The connection with the frequency of child punishment in the family, in our opinion, is better explained by the subscales of the neuroticism scale and the trust subscale. These personality traits may affect the frequency of child punishment in the family through emotional stress. The data obtained allow us to outline further studies of personality traits in connection with the practice of punishing children in the family and to come to better understanding the influence of the neuroticism factor.

Personal characteristics of the parents affect the frequency of punishments in the family, both directly, through the behavioral characteristics of personality traits, and indirectly, through satisfaction with their family life and less exposure to emotional stress. Correlation indicators by the VEC test and the California Psychological Inventory (CPI) help to explain the results.

VEC correlations with CPI scales, also obtained in our previous studies, are as follows. The strongest correlation is found between VEC and the dominance CPI scale (r = 0.54, *p* < 0.01). Positive VEC correlations are available with the scales for capacity for status (r = 0.25, *p* < 0.05), sociability (r = 0.38, *p* < 0.01), social presence (r = 0.29, *p* < 0.01), self-acceptance (r = 0.35, *p* < 0.01), sense of well-being (r = 0.22, *p* < 0.05), good impression (r = 0.27, *p* < 0.05), achievement via conformance (r = 0.26, *p* < 0.05), psychological-mindedness (r = 0.21, *p* < 0.05), and flexibility (r = −0.27, *p* < 0.05) [12]. The correlation with the CPI can be used to describe a person with a high VEC score. This is a person with the following traits: Leadership, social initiative (Do); a desire to make a career and efficiency in (Cs); sociability and openness (Sy); spontaneity in communication, self-confidence (Sp); self-esteem (Sa); not emphasizing troubles and discontent (Wb); a desire to make a good impression (Gi); an ability to adjust to the social environment (Ac); an ability to respond to the needs and experiences of other people (Py); a rigid respect for customs and traditions (Fx). These personality traits may be associated with communication help to increase satisfaction with family life, such as satisfaction with how problems are discussed and decisions are made in the family. The data obtained allow us to outline further studies of personality traits considering the practice of punishing children in the family and to study more thoroughly the influence of factors related to communication between spouses. Different types of punishments were considered in the research.

To analyze the results obtained in the study, it is necessary to consider what types of punishments are reflected in the scales as well as the situations of the child’s behavior; in relation to both elements, the subjects gave a self-report on the frequency of punishment. The child’s behavioral reactions, in relation to which parents had to assess the frequency of punishment in the second part of the diagnostic scales of punishment rates for children, are as follows: (1) Brother and sister (ages four and seven) quarrel over a book. The parent intervenes and decides that the youngest child has the right to play with this book first; then the eldest may play with the book. The older child throws a tantrum and tears the book in front of the eyes of his parents. (2) As a punishment for committing this action, the parent spanks the child who is 10 years old. He tries to answer the parent in kind. (3) The parent, as a punishment for such impudence, makes the child (12 years old) help with the cleaning. The child answers back. (4) For many weeks, a child (nine years old) does his lessons irregularly and carelessly. One day, he returns home with a weekly school progress report full of bad grades. (5) The supervising teacher of a child (11 years old) calls the parents of the child and complains that the child does not obey, is impudent, and disrupts the work of the class. (6) A child (seven years old) stubbornly refuses to eat anything of what the child’s parents have prepared for lunch. (7) Without any good reason, a child (12 years old) returned home two hours late. (8) The parent found that the child stole 100 rubles from his wallet. The research objectives did not include pedagogical assessment of the justification and legitimacy of the punishment application in any particular situation. The subject matter of analysis was the frequency of punishments. However, not all situations are equivalent. For example, the second situation, to a lesser extent, actualizes the use of punishment because the parent provoked the child’s reaction with his own behavior. The types of punishment and their frequency must be assessed in eight situations as follows: (1) To prohibit watching TV for some time. (2) To make it clear to the child that he has upset his parents. (3) To give the child a slap upside the head. (4) To put the child in the corner for a while and do not pay attention to him. (5) To cut back the child’s pocket money. (6) To spank the child. (7) To tell the child that next time he will receive severe punishment. (8) To make the child apologize. (9) To deprive the child of anything that he wants and several activities which usually he is allowed to do (excursion, playing with a toy, etc.). (10) To forbid the child to play with his friends (male or female) after lunch. The scale assesses the parent behavior in an imaginary situation; this refers to expected behavior, about a tendency to behave in a certain way, and not about real behavior. Despite the ethical and pedagogical expediency of certain punishments is doubtful, the tendency of parents to apply these types of punishments is of scientific interest. The tendency to resort to punishments 3 and 6 is interpreted by us as a negative trend in parental behavior. This is a behavior that can harm the child. We consider the fact noted by parents in their self-reports about such punishment application to constitute additional evidence of the sincerity of their answers. The punishments that consist in the prohibition of an activity or the deprivation of an item usually allowed to the child to be a response to the child’s socially unacceptable behavior, impeding the formation of this undesirable behavior in the child; these punishments are interpreted by us as a necessary element of education. We consider these punishments as a negative trend in the interaction with a child only when the frequency of their application is excessive. Some punishments from the above list are not only the markers of undesirable behavior but they also form socially desirable behavior. For example, a requirement for a child to apologize when he is wrong. In this case, this largely constitutes negative reinforcement rather than punishment. If a parent makes it clear to the child that the child’s behavior is distressing, it is not so much about punishment as helping the child learn how to navigate social relationships. Thus, a child learns to understand that his socially negative behavior upsets people close to him. Such types of parental behavior are interpreted in our study not so much as punishments but as educational practices that shape certain types of behavior. Nevertheless, the requirement to apologize and show the child that he has upset his parents is seen by many parents as punishment. The obtained negative correlations with the frequency of punishments included in this scale indicate that the higher the volitional traits and emotional stability of parents are expressed, the more often they punish children. These data do not contradict the general logic of the results obtained in the study, since all correlations reflect the frequency of applying such types of punishments as follows: let the child understand that he has upset his parents (2) and demand an apology (8). Parents resort to this type of behavior in situations 1, 4, 5, and 7. That is, in situations where the child does not control his anger, neglects school responsibilities, violates discipline at school, and returns home late. The data obtained indicate reasonable and sound educational practices of parents with high VEC indexes.

As shown in the Methods section, the scale was used to diagnose the frequency of punishments, in which parents gave a self-report on applying 12 types of punishments in relation to their children. This scale refers to the real behavior of parents. The parents give a self-report on how often they apply a specific type of punishment when the child misbehaves. In this case, none of the specific situations of the child’s behavior were given. This scale contains either prohibitions or the behavior that we interpret as illegitimate in relation to the child. To deprive a child of a dessert is an example of an activities prohibition that is used as a punishment in this scale. A punishment such as spanking a child is an example of behavior that could have negative consequences for a child. High rates on this scale indicates frequent application of punishments and is interpreted as a negative trend in the interaction of parents and their child. The differences between groups, although not reaching statistical significance, relate to this scale. A group of parents with high VEC scores are less likely to punish their children.

A positive correlation was found between the frequency of punishments applied by parents and the frequency of the same punishments applied to these parents in their own childhood by their parents, indicating the importance of parental behavior patterns. Since these behaviors relate to the practice of punishment, they are most likely related to the negative emotions of the parents in their childhood. Nevertheless, they were perceived and proved to be stable and applicable in behavior in relation to their own children.

The data in our research show what traits of a parent’s personality are significant in terms of the frequency of child punishment in the family. Further research in this vein may allow us to determine the type of personality that contributes to low frequency of punishments, to the choice of optimal educational strategies, the type of personality associated with high frequency of punishments, and with the choice of not optimal types of punishments.

The interconnection between depressed mood in family life or the stress caused by the child-rearing responsibilities and volitional traits does not reach statistical significance and can be interpreted as a tendency coinciding with the common laws revealed in our research.

## 5. Conclusions

The more expressed personal and behavioral qualities such as self-confidence, independence, the ability to control anger, activity, perseverance, and the presence of clear goals in life, the higher the satisfaction with marriage, the lower the severity of emotional stress, and the frequency of punishment of children in the family.

The relationship between the volitional and emotional personality traits diagnosed with the VEC and satisfaction with family life and emotional stress are consistent. They allow for the diagnosis of personal characteristics in parents that reduce the likelihood of frequent punishment of children in the family.

The volitional traits and emotional stability of the person indirectly influence the frequency of punishments in the family. Family life satisfaction and stress levels in the family, influenced by the parents’ personal traits, are connected with the frequency of punishments.

The research is limited only by the small number of personal characteristics of parents and the punishments of children in the family. In future research, it is possible to expand the number of personal characteristics and consider the relationship not only with punishments but also with rewards.

## Data Availability

The datasets used and analyzed during the current study are available from the corresponding author on reasonable request.

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
