# Peer review of "Parents’ Personality, Marriage Satisfaction, Stress, and Punishment of Children in the Family"

_behavsci, 2019, doi:10.3390/bs9120153_

Round 1

Reviewer 1 Report

The paper examines parents’ personality, marriage satisfaction, stress, and punishment of children in the family. It deals with an interesting topic that is both novel and important. The literature is also adequate and the methods and reporting procedures are largely appropriate.

However, the discussion section is a very lengthy section, which I would suggest segmenting into separate, meaningful sub-sections. For example, the authors can consider discussing the general findings based on the data analysis results in the discussion section, and following that, to discuss its implications for theory and practice in separate sub-sections labelled as “theoretical implications”, “managerial implications”, and “policy implications” (optional). Moreover, the conclusion section needs to be elaborated, specifically in terms of the study’s limitations and directions for future research.

Finally, two minor suggestions.

First, the sentence “analysis of variance (ANOVA) was applied to compare groups” can be cited with the following reference:

Lim, W. M. & Ting, D. H.. (2012). A toolkit of sampling and data analysis techniques for quantitative research. Munich, Germany: GRIN Publishing.

Second, the sentence “the groups are homogeneous” can be expanded to “the groups are homogeneous, a condition suitable for experimental research” and can be cited with the following references:

Lim, W. M. (2015). Enriching information science research through chronic disposition and situational priming: A short note for future research. Journal of Information Science, 41(3), 399-402.

Lim, W. M., Ahmed, P. K., & Ali, M. Y. (2019). Data and resource maximization in business-to-business marketing experiments: Methodological insights from data partitioning. Industrial Marketing Management, 76, 136-143.

I hope the authors will find these comments useful to improve their paper. All the best!

Author Response

Please see attachment. Thanks for the comments, that help improve the article!

Reviewer 2 Report

The pourpose of the presen study was  to ascertain whether emotional and volitional characteristics of the individual affect the frequency of punishment in the family. As a results, authors suggested that the emotional and volitional qualities of the parents' personality indirectly influence the frequency of punishment of children in the family.

This study need to be revised several points.

1) Authors mentioned that 'The hypothesis of the interconnection between volition an emotional stability and low frequency of punishment of children was to be verified.' However, this need to be revised more exactly.

2) What is the ethical committee approved no. in the present study?

3) In the Materials and Methods, I think that this is very dificult to understand for readers, so could you revise methods of stastical analysis more exactly?

4) There were no figure and tables in the present study, so this need to be revised and/or added in the results.

Author Response

Please see attachment. Thank you for your remarks!

Round 2

Reviewer 2 Report

Thanks for your revision besed on the reviwers comments.

This manuscript will be able to publish as soon as possible.